# Ventilator-Associated Lung Injury: Pathophysiology, Prevention, and Emerging Therapeutic Strategies

**DOI:** 10.3390/ijms262110448

**Published:** 2025-10-28

**Authors:** Ana Costa, Bintia Sakho, Sangel Gomez, Brandon Khanyan, Pamella Leybengrub, Sergio Bergese

**Affiliations:** 1Department of Anesthesiology, Renaissance School of Medicine, Stony Brook University, Stony Brook, NY 11794, USA; ana.costa@stonybrookmedicine.edu (A.C.);; 2Renaissance School of Medicine, Stony Brook University, Stony Brook, NY 11794, USApamella.leybengrub@stonybrookmedicine.edu (P.L.)

**Keywords:** lung injury, ventilator-associated lung injury, ventilator-induced lung injury, personalized medicine, acute respiratory distress syndrome, mechanical ventilation

## Abstract

Mechanical ventilation is a critical intervention in patients who cannot spontaneously maintain adequate oxygenation and remove carbon dioxide. However, it can also lead to severe lung injury via volutrauma, barotrauma, atelectrauma and biotrauma, and it can worsen existing lung disease such as acute respiratory distress syndrome. Ventilator-associated lung injury, the clinical manifestations of lung damage associated with mechanical ventilation, can trigger systemic inflammatory cascades that contribute to multi-organ failure. The utilization of lung-protective ventilation strategies helps to minimize further injury to the lungs during mechanical ventilation and improve survival rates. This review discusses the pathophysiology of ventilator-associated lung injury, including cellular and molecular responses, its systemic effects, risk factors, clinical presentation and diagnosis, protective strategies, and emerging therapies. It incorporates interdisciplinary advances, from novel pharmacologic and stem-cell therapies coupled with artificial intelligence and machine learning systems to provide a framework for the prevention of ventilator-associated lung injury that moves beyond purely mechanical considerations.

## 1. Introduction

Mechanical ventilation is a lifesaving intervention when spontaneous ventilation is compromised such as in trauma, neuromuscular weakness, sepsis, respiratory failure, etc. It provides a mechanism for oxygenation and carbon dioxide removal. However, mechanical ventilation can cause lung injury via volutrauma, barotrauma, atelectrauma and biotrauma, and it can exacerbate conditions including acute respiratory distress syndrome (ARDS), as well as increasing morbidity and mortality in intensive-care patients. Thus, in patients where mechanical ventilation is a lifesaving intervention, lung protective ventilation strategies must be utilized to minimize harm.

Ventilator-associated lung injury (VALI) refers to the clinical manifestations of lung damage associated with mechanical ventilation [1]. Ventilator-induced lung injury (VILI) refers to the actual biological damage to the lung caused directly by mechanical ventilation. VILI is often morphologically and physiologically indistinguishable from VALI, and it can only be discerned definitively in animal models [2]. VILI is sometimes used as a synonym for VALI, but VILI is a term better applied to a patient with VALI where mechanical ventilation is the proven cause of the lung injury. On the other hand, VALI is a broader term utilized when the cause of lung injury has not been determined and may have been caused by mechanical ventilation or worsening of underlying lung conditions while on mechanical ventilation.

The exact incidence of VALI is difficult to determine since it overlaps with ARDS and other causes of lung injury, and it is difficult to establish causation of lung injury by mechanical ventilation alone. Gajic et al. found that 24% of the 332 mechanically ventilated patients without acute lung injury at the onset developed new lung injury by day 5 [3]. The LUNG SAFE study demonstrated that 10.4% of intensive care unit (ICU) admissions and up to 23% of mechanically ventilated patients met criteria for ARDS, and VALI may contribute to or worsen the condition [4]. In a cohort of 1113 mechanically ventilated patients, Rubenfeld et al. found the incidence of acute lung injury was 78.9 cases per 10,000 person-years, the incidence of ARDS was 58.7% cases per 100,000 person-years, and the in-hospital mortality rate for patients with acute lung injury was 38.5% whereas the mortality rate for patient with ARDS was 41.1% [5].

Since mechanical ventilation is a lifesaving measure but can also cause severe injury, it is critical to understand VALI in order to optimize mechanical ventilation settings and minimize further injury to critically ill patients. Furthermore, VALI can trigger systemic inflammatory cascades that contribute to multi-organ failure [6]. The ARDSNet Low Tidal Volume Trial demonstrated significantly lower mortality with lower tidal volumes in ARDS ventilated patients compared to traditional tidal volume ventilation (31% vs. 39.8%) [7]. Thus, preventing VALI via lung-protective ventilation contributes to better survival rates and decreased length of ICU stays. This review discusses the pathophysiology of VALI, including cellular and molecular responses, its systemic effects, risk factors, clinical presentation and diagnosis, protective strategies, and emerging therapies (Figure 1).

## 2. Pathophysiology of Ventilator-Associated Lung Injury

### 2.1. Mechanisms of Injury

VALI occurs when the forces from mechanical ventilation are greater than what the lung tissue can tolerate. There are four main ways this damage occurs: barotrauma, volutrauma, atelectrauma, and biotrauma [8]. Although these processes are often described individually, the effects are compounded by one another [9].

Barotrauma results from alveolar pressures that exceed the structural limits of the lung, producing direct mechanical disruption of the alveolar–capillary interface. The most recognizable consequences are pneumothorax, pneumomediastinum, and interstitial emphysema [10]. Although current practice is to limit plateau pressures to below 30 cm H_2_O, this threshold must be studied further [11]. Additionally, some regions of the lung may still be exposed to damaging pressures even when the ventilator parameters are within the recommended ranges [10].

In contrast, volutrauma refers to alveolar overdistension due to large tidal volumes. Stress and strain are terms that describe the effect of external forces that act on a subject. Lung stress is the distribution of forces per unit of area due to tidal volume and positive end-expiratory pressure (PEEP), and the alteration in lung volume as a result is defined as strain [12]. Too much stress and strain can lead to lung injury by causing local and systemic inflammation [13,14]. High tidal volumes and inspiratory pressures, elevated driving pressure, and inappropriate PEEP settings can all lead to excessive stress and strain, resulting in volutrauma and barotrauma [14,15]. A retrospective study examined the relationship between driving pressure and lung stress in patients with ARDS, and found that at both a PEEP of 5 and 15 cm H20, the higher driving pressure group had a significantly higher lung stress [16]. A secondary analysis of 1705 mechanically ventilated patients found that driving pressure was also independently correlated with mortality [adjusted OR, 1.04 (1.01–1.07)] [17]. Driving-pressure-guided ventilation is a method aimed at protecting the lungs from damage. A meta-analysis of seven randomized controlled trials demonstrated that patients under driving pressure-guided ventilation had decreased mortality compared to the higher driving pressure control group (RR 0.56; 95% confidence interval [CI], 0.39–0.79; *p* = 0.001; I2 = 23%). PaO2/FiO2 was also statistically significantly higher in the driving-pressure-guided group than the control, indicating this strategy can lower mortality and improve oxygenation [18]. Overdistension disrupts epithelial and endothelial barriers, increases capillary leak, and results in pulmonary edema. Experimental methods have shown that these changes can occur even without elevated plateau pressures, suggesting that volume may be an independent cause of injury [10]. Clinically the ARDSnet trial demonstrated that using lower tidal volumes (6 mL/kg predicted body weight) could significantly reduce mortality [7]. However, the specific lower limit of safe tidal volume remains up to debate, especially in patients without ARDS.

Atelectrauma describes the mechanical strain caused by repetitive alveolar collapse and reopening, often occurring in poorly recruited lung units. This cyclic opening and closing creates shear stress on epithelial surfaces, promoting edema and worsening gas exchange [19]. Furthermore, since surfactant function is impaired during mechanical ventilation, the risk of collapse of peripheral bronchioles at low lung volumes increases [20]. The resulting alveolar instability promotes uneven stress distribution and localized lung injury.

Biotrauma extends beyond structural injury to include the biological response triggered by mechanical forces. Overdistention and shear forces activate intracellular pathways resulting in the release of inflammatory mediators which amplify local inflammation and can disseminate systemically [21]. Although the concept of biotrauma has been demonstrated in both animal and human studies, the extent of its clinical effect remains disputed. Nonetheless, it highlights the physiological pathways that link mechanical ventilation to systemic disease processes.

### 2.2. Cellular and Molecular Responses

Among the many inflammatory mediators, cytokines such as interleukin-1β (IL-1β), interleukin-6 (IL-6), interleukin-8 (IL-8) and tumor necrosis factor-α (TNF-α) are consistently elevated in ventilator-induced injury models [22]. They have also been found to upregulate adhesion molecules, including E-selectins, P-selectins and ICAM-1, and initiate signaling pathways such as NF-κB and MAPK [23]. The resulting increase in alveolar-capillary permeability via cytokine-induced damage worsens pulmonary edema and gas exchange efficiency. Endogenous molecules have also been found to potentiate inflammation by activating pattern recognition receptors and cytokine release in response to lung tissue injury. Studies in both animals and humans have confirmed this process, showing elevated damage-associated molecular patterns in bronchoalveolar lavage fluid of patients with VILI and increased expression of pattern recognition receptors in ventilated lung tissue [24].

Oxidative stress also plays a key role in the development of VALI. Mechanical stretch and inflammatory signaling promote the release of reactive oxygen species (ROS) from alveolar macrophages and recruited neutrophils [23]. These ROS damage lipids, proteins, and deoxyribonucleic acid, while also activating transcription factors that increase cytokine production [23]. While there are natural antioxidant defenses such as glutathione, they are often overwhelmed in the setting of critical illness and prolonged ventilation.

The production of cytokines and ROS relies on the involvement of immune cells, predominantly neutrophils and macrophages. As explained, recruited neutrophils release proteases and oxidants, which disrupt alveolar architecture. Macrophages play a dual role, producing inflammatory cytokines in the acute phase, but later shifting toward tissue repair which may promote fibrotic remodeling [21]. Other immune populations including eosinophils and Th2 lymphocytes have also been linked to the release of cytokines [23].

### 2.3. Systemic Effects

Although initially a pulmonary process, VALI frequently extends beyond the lungs. Circulating cytokines, chemokines, and extracellular vesicles released from injured alveoli contribute to multiorgan dysfunction through various mechanisms, including systemic endocrine responses [25]. This systemic involvement has been linked to acute kidney injury, hepatic dysfunction, and cardiovascular compromise in both animal and clinical studies [26]. In addition, growing evidence highlights the connection between the coagulation cascade and inflammation in VALI. In primate models of sepsis-induced acute lung injury, targeting the intrinsic coagulation pathway reduced fibrin deposition, attenuated inflammation, and improved lung function [21].

In patients with preexisting systemic inflammation, such as from sepsis or systemic inflammatory response syndrome, VALI may be especially dangerous. Ventilation can exacerbate inflammation in septic patients by amplifying cytokine release, while in turn, systemic inflammation lowers the threshold for ventilator-induced damage [27]. This bidirectional relationship helps explain the individualized risks and disproportionately high morbidity and mortality observed in septic patients requiring mechanical ventilation (Figure 2).

## 3. Risk Factors and Patient Susceptibility

Underlying lung diseases, such as ARDS and chronic obstructive pulmonary disease (COPD), have been found to increase a patient’s risk of VALI. The ARDS lung is compared to “a small baby lung” due to the reduction in the number of functional alveoli during breaths. Ventilation with large tidal volumes has demonstrated an increase in morbidity and mortality in ARDS patients due to induced volutrauma [21]. Furthermore, the variability of inflammatory change within the lungs in ARDS patients acts as a stressor that predisposes patients to lung injury. While positive end-expiratory pressure (PEEP) is needed to promote opening of alveoli, high PEEP levels can worsen lung stress and tension leading to VALI [28]. Furthermore, although efforts have been made to reduce tidal volume and inspiratory pressures, mortality from lung injury in patients with ARDS remains elevated [29].

Similarly to ARDS, COPD and other obstructive diseases including asthma have mechanical heterogeneity due to regional variability in factors such as bronchoconstriction and airway secretions. This makes patients with these lung diseases more vulnerable to lung injury [30]. COPD patients can have diaphragm and skeletal muscle weakness due to corticosteroid use and inflammation, increasing their length of duration on mechanical ventilation and putting them at greater risk for VALI [31]. Thus, underlying lung diseases such as COPD and ARDS increase patient susceptibility to VALI.

VALI is also impacted by several other risk factors. Older age and comorbidities, such as pneumonia and immunosuppression, are risk factors for risk and severity of VALI [32]. There are also numerous ventilation parameters that predispose patients to VALI. High PEEP can lead to overdistention of the lung and hemodynamic impairment. High inspiratory flow rates can cause microvascular injury and diminish gas exchange. Increased inspiratory time has been found to reduce compliance and oxygenation and high inspiratory frequency increased perivascular hemorrhage in animal models [8]. Plateau and peak pressures and extended duration of mechanical ventilation have also been found to increase risk of VALI [29,33]. Neutrophil gelatinase-associated lipocalin, a key marker in the innate immunologic response and inflammation, was found to be elevated in a time-dependent manner under mechanical ventilation [34].

## 4. Clinical Presentation and Diagnosis

### 4.1. Clinical Signs and Symptoms

Understanding the clinical signs and symptoms of VALI requires a foundational knowledge of its underlying pathophysiology. A key initial event in VALI is the acute inflammation of the alveolar-capillary membrane. This inflammation compromises the integrity of the barrier, resulting in the extravasation of protein-rich fluid into the alveolar spaces and interstitium [35]. The vascular component in the development of pulmonary edema is twofold. During lung inflation, the pressure in the interstitium surrounding the alveolar blood vessels decreases, while the transmural pressure within these vessels increases. This pressure differential promotes vascular dilation and leakage [9]. In addition, surfactant dysfunction contributes to alveolar collapse and reduced lung compliance, further exacerbating gas exchange abnormalities.

The hallmark clinical manifestation of VALI is pulmonary edema accompanied by impaired oxygenation. This is typically represented by hypoxemia, as reflected by a decline in arterial oxygen partial pressure, necessitating increased fractional inspired oxygen levels. Impaired alveolar ventilation may lead to the retention of carbon dioxide, resulting in respiratory acidosis. Conversely, in the early phases of lung injury, hyperventilation may cause hypocapnia.

Patients with VALI often present with tachypnea, the use of accessory respiratory muscles, and tachycardia [36]. However, these signs can be difficult to assess in mechanically ventilated patients. Furthermore, administration of high concentrations of oxygen can result in hyperoxia, which may induce oxidative alveolar injury [35]. Decreased lung compliance and/or increased airway resistance contribute to elevated peak and plateau airway pressures. Additional complications associated with VALI, particularly due to barotrauma, include pneumothorax, pneumomediastinum, and subcutaneous emphysema [9].

### 4.2. Radiographic Findings

Chest radiography in patients with VALI typically reveals bilateral alveolar and/or interstitial infiltrates. These infiltrates are often diffuse and may produce a characteristic “white lung” appearance. Importantly, a normal cardiac silhouette can help distinguish VALI from cardiogenic pulmonary edema [37]. Computed tomography imaging further elucidates the extent of lung injury. Common findings include patchy, heterogeneous areas of consolidation and atelectasis. Additionally, regions of hyperlucency may occasionally be observed, indicating areas of overdistension or air trapping.

### 4.3. Biomarkers

Mechanical stress imposed on the lungs during mechanical ventilation can activate inflammatory responses even in the absence of overt structural damage. This process, known as biotrauma, involves the release of pro-inflammatory cytokines, such as TNF-α and IL-6, as well as the recruitment of neutrophils and induction of oxidative stress. These events are mediated by mechanotransduction, a process whereby mechanical forces trigger intracellular signaling pathways, ultimately resulting in cellular activation and injury [35].

An effective biomarker would ideally quantify the extent of lung injury and guide therapeutic interventions. Elevated systemic levels of inflammatory markers such as TNF-α, IL-6, and IL-8 have been associated with adverse outcomes in patients with ARDS. However, these markers are not specific to VALI [35]. Recent investigations have explored more specific biomarkers for alveolar epithelial damage. The receptor for advanced glycation end products (RAGE), a marker of alveolar type I cell injury, has shown promise in this regard. Baseline plasma levels of RAGE correlate with the severity of lung injury, and elevated levels have been associated with worse clinical outcomes in patients with acute lung injury. In a study by Calfee et al., higher plasma RAGE concentrations were observed in patients ventilated with higher tidal volumes, while those ventilated with lower tidal volumes demonstrated a more significant decline in RAGE levels, suggesting a potential role for RAGE in monitoring ventilator-induced injury [38].

Similarly, surfactant protein D (SP-D), derived from alveolar type II cells, has also been proposed as a biomarker. Elevated plasma levels of SP-D have been linked to worse outcomes in lung injury. Importantly, patients ventilated with lower tidal volumes exhibited significantly lower SP-D levels, further supporting the protective effect of lung-protective ventilation strategies [39]. Although these biomarkers offer valuable insights into lung injury mechanisms, none are currently specific to VALI, limiting their utility in distinguishing VALI from other forms of lung inflammation or injury.

### 4.4. Differentiating Ventilator-Associated Lung Injury from Disease Progression

VALI often presents with clinical features that closely resemble progressive pulmonary conditions such as ARDS, including refractory hypoxemia, diffuse bilateral infiltrates on imaging, and reduced lung compliance. Despite these similarities, the underlying pathophysiological mechanisms differ, making accurate differentiation essential for appropriate clinical management. VALI results from mechanical forces imposed by the ventilator, particularly barotrauma and volutrauma, which arises from alveolar overdistension associated with high tidal volumes. These processes disrupt the alveolar-capillary membrane, leading to increased permeability and the extravasation of protein-rich fluid into the alveoli [40]. Elevated transpulmonary pressures further exacerbate lung injury by amplifying mechanical stress, thereby promoting inflammation, pulmonary edema, and impaired gas exchange [10]. In contrast, disease progression in ARDS is primarily driven by intrinsic inflammatory or infectious etiologies, which may worsen despite lung-protective ventilation strategies [41]. Differentiating VALI from ARDS progression thus requires integrating ventilator parameters, clinical trajectory, imaging findings, and the patient’s physiological response to supportive care.

## 5. Prevention and Protective Strategies

### 5.1. Lung-Protective Ventilation

ARDSNet, one of the most pivotal studies in the management of ARDS was conducted by the ARDS Network and published in 2000. This multicenter, randomized controlled trial enrolled 861 patients with ARDS, comparing the effects of traditional ventilation strategies using a tidal volume of 12 mL/kg of predicted body weight and plateau pressures ≤ 50 cm H_2_O versus a lung-protective strategy using a tidal volume of 6 mL/kg and plateau pressures ≤ 30 cm H_2_O [7]. The findings demonstrated a significant reduction in mortality with the low-tidal-volume strategy, showing an absolute survival benefit of approximately 9%. Additionally, patients in the low-tidal-volume group experienced shorter durations of mechanical ventilation and had lower levels of circulating inflammatory cytokines, suggesting a reduction in VILI [7]. Overall, these findings established low tidal volume ventilation as a fundamental strategy for reducing mortality and ventilator days in patients with acute lung injury and ARDS.

Lung-protective ventilation strategies are central to the management of ARDS, with low tidal volume ventilation and appropriate PEEP application forming the foundation of evidence-based practice. PEEP plays a critical role in improving oxygenation and minimizing atelectrauma. However, the optimal level of PEEP has been the subject of extensive investigation. Early randomized controlled trials comparing higher versus lower PEEP levels demonstrated improvements in oxygenation but failed to show a significant mortality benefit [42]. Building on these observations, subsequent research explored more comprehensive ventilation protocols. Meade et al. investigated a multifaceted ventilatory approach incorporating low tidal volumes (6 mL/kg predicted body weight), plateau pressures limited to ≤40 cm H_2_O, recruitment maneuvers, and higher PEEP to maintain an “open-lung” strategy. This approach was associated with improved oxygenation and a reduced need for rescue therapies, though it did not translate into a clear survival advantage [43]. Similarly, Mercat et al. tested the hypothesis that titrating PEEP to achieve higher alveolar recruitment could yield clinical benefits. While this strategy improved lung mechanics, enhanced oxygenation, and shortened the duration of mechanical ventilation, it did not significantly reduce mortality compared to conventional PEEP strategies [44].

Overall, these studies suggest that while higher PEEP may improve physiological parameters and reduce ventilator dependency, it does not consistently confer a survival advantage. Thus, the choice of PEEP should be individualized, balancing the potential benefits of alveolar recruitment against the risks of overdistension and hemodynamic compromise.

### 5.2. Adjunctive Therapies

Prone positioning is an adjunctive therapy that helps to reduce VALI through several mechanisms. By redistributing transpulmonary pressures more uniformly, it lessens regional stress and strain, enhances ventilation–perfusion matching, and limits overdistension in anterior lung regions [35]. Initial randomized controlled trials demonstrated improvements in oxygenation with prone positioning but did not consistently translate these physiological benefits into reduced mortality when compared with supine ventilation. A systematic review by Sud et al. reinforced this pattern, noting better oxygenation and a lower incidence of ventilator-associated pneumonia, though survival benefits remained uncertain [45]. The landmark PROSEVA trial altered this perspective by enrolling patients with severe ARDS and implementing early, prolonged prone sessions. In contrast to earlier studies, PROSEVA reported substantial reductions in both 28-day and 90-day mortality among patients managed in the prone position compared with supine ventilation [46]. These improvements were achieved without an increase in serious adverse events, although complications such as pressure injuries and airway-related issues remain possible. Current evidence therefore supports prone positioning as a standard component of care in severe ARDS, rather than a last-resort measure. Nonetheless, its specific role in attenuating VALI continues to warrant further study.

Patient–ventilator asynchrony has been associated with prolonged mechanical ventilation and worse clinical outcomes. Neuromuscular blocking agents (NMBAs) may help mitigate this effect by improving patient–ventilator synchrony. In a multicenter, randomized, double-blind trial involving 340 patients with severe ARDS, participants were assigned to receive either a 48 h infusion of cisatracurium besylate, a neuromuscular blocking agent, or placebo. The study demonstrated an absolute reduction in 28-day mortality of 9.6% in the NMBA group compared with the control group (*p* = 0.05), indicating significantly improved survival at both 28 and 90 days without an increased risk of ICU-acquired paresis [47]. Lung-protective ventilation strategies were applied in both groups. Additionally, patients receiving NMBAs experienced fewer days on mechanical ventilation, lower incidence of pneumothorax, reduced duration of ICU stay, and decreased occurrence of barotrauma [47]. These findings suggest that neuromuscular blockade may play a beneficial role in reducing VALI, but further studies are warranted to clarify optimal patient selection, timing, and duration of NMBA therapy to maximize clinical benefit while minimizing adverse effects.

Atelectasis is a common complication in mechanically ventilated patients and can be mitigated through the use of recruitment maneuvers. Recruitment maneuvers involve intermittent or sustained hyperinflation of the lungs to reopen collapsed alveoli, often combined with elevated PEEP to prevent re-collapse. These interventions serve as valuable adjuncts to low tidal volume ventilation, helping to reduce the risk of atelectrauma. The most frequently employed recruitment maneuver is sustained inflation. However, this approach carries potential adverse effects, including barotrauma, alveolar overdistention, and hemodynamic instability. Recent strategies, such as pressure-controlled ventilation with gradual or incremental PEEP increases, have been proposed to minimize these risks [48].

In a large randomized controlled trial comparing conventional low-tidal-volume ventilation to an “open lung” approach—which combined low tidal volume, higher PEEP, and recruitment maneuvers—the addition of recruitment maneuvers did not significantly affect overall mortality, duration of mechanical ventilation, or incidence of barotrauma. Nevertheless, the open-lung strategy was associated with reduced need for rescue therapies and fewer deaths due to refractory hypoxemia [43]. Current evidence supports the use of recruitment maneuvers primarily as a method to optimize PEEP settings or as a rescue intervention in cases of severe hypoxemia, rather than as a routine, standalone procedure [48]. Consequently, while recruitment maneuvers can be a beneficial adjunct in selected patients with severe hypoxemia, their routine application is limited by the potential for barotrauma and cardiovascular complications.

### 5.3. Non-Ventilatory Strategies

Pulmonary edema represents a major component of VALI, highlighting the critical role of fluid management in modulating its development and progression. Evidence suggests that judicious fluid management can reduce the incidence of pulmonary edema and increase the number of ventilator-free days [49]. The Fluid and Catheter Treatment Trial (FACTT) further demonstrated that a conservative fluid management strategy in patients with acute lung injury and ARDS significantly increased ventilator-free (14.6 ± 0.5 vs. 12.1 ± 0.5, *p* < 0.001) and ICU-free days (13.4 ± 0.4 vs. 11.2 ± 0.4, *p* < 0.001) with improved lung function [50]. This randomized controlled trial consisted of 1000 patients with acute lung injury on either a conservative or liberal fluid management protocol for seven days. However, inadequate fluid resuscitation in mechanically ventilated patients may lead to hypovolemia, which in turn can precipitate multiple organ failure [6]. Furthermore, research indicates that age-related susceptibility influences the severity of VALI-induced pulmonary edema. Older patients are not only more prone to injury but are also more likely to experience downstream complications and increased mortality [51]. These findings underscore the importance of achieving an optimal balance in fluid management to improve patient outcomes and reduce the risk of ventilator-associated complications.

Although there is limited literature regarding early weaning protocols and sedation minimization in VALI, emerging evidence supports their potential benefits in mechanically ventilated patients. A clinical trial involving 104 adults investigated whether the early initiation of physical and occupational therapy, combined with daily sedation interruptions, could improve recovery outcomes in the ICU. The study found that 59% of patients in the early therapy group regained independent functional status—defined as the ability to walk unassisted and perform six activities of daily living—by hospital discharge, compared with 35% in the control group (*p* = 0.02; odds ratio 2.7, 95% CI 1.2–6.1). Moreover, patients in the intervention group experienced a greater number of ventilator-free days within the first 28 days (median 23.5 days vs. 21.1 days, *p* = 0.05) [52]. These findings indicate that the early implementation of structured mobilization alongside sedation minimization is a safe and effective strategy to enhance both physical and neurocognitive recovery in critically ill patients. A summary of key interventions and their demonstrated effects are shown in Table 1.

## 6. Emerging Therapies and Research

### 6.1. Personalized Ventilation Strategies

Traditional lung-protective strategies, such as low tidal volumes and limiting plateau pressures, have improved outcomes in ARDS, yet mortality remains high. To further advance care, researchers are exploring personalized mechanical ventilation tailored to each patient’s physiology and lung morphology. The LIVE study demonstrated that aligning tidal volumes, PEEP, prone positioning, and recruitment with “focal” and “non-focal” ARDS subphenotypes could reduce mortality when morphology was accurately classified [53]. However, benefits were obscured by frequent misclassification. The PEGASUS trial (Personalized Mechanical Ventilation Guided by Ultrasound in ARDS) will be the first randomized controlled trial to evaluate lung ultrasound–guided personalized ventilation compared with conventional strategies in moderate to severe ARDS [54]. It plans to enroll 538 patients that are mechanically ventilated in an ICU setting with moderate to severe ARDS.

Additional targets of ventilator personalization are under investigation. Transpulmonary pressure, derived from airway and pleural pressure estimates, reflects the lung’s true distending force and can guide individualized adjustment of PEEP, inspiratory limits, and recruitment maneuvers, though robust trial evidence remains limited [55]. Mechanical power, representing the cumulative energy delivered from the ventilator to the lungs, integrates tidal volume, pressure, flow, resistance, and respiratory rate. Multiple studies have demonstrated that higher mechanical power is associated with VILI in various patient populations, including critically ill patients and even low-risk surgical patients [17,56,57,58,59]. One study looking at critically ill patients admitted to intensive care units in hospitals across the United States demonstrated an association between high mechanical power and ICU-related hospital mortality [56]. A secondary analysis of 1705 mechanically ventilated patients found an association between mechanical pressure and development of ARDS (adjusted OR, 1.03) [17]. Similarly, a multi-center retrospective cohort study of two hospital systems in Boston found a statistically significant association between higher mechanical power and risk of developing postoperative respiratory failure [adjusted odds ratio, 1.31 per 5 J/min increase; 95% CI, 1.21 to 1.42; *p* < 0.001] [57]. Limiting mechanical power can reduce the disruption of cells and the extracellular matrix resulting from sustained high-energy stress [60]. Thus, restricting mechanical power has been proposed to limit VILI, as exceeding critical thresholds accelerates damage. However, uncertainty persists regarding calculation methods, validated thresholds, and randomized controlled trial confirmation [29]. Despite promising concepts, meta-analyses show no significant mortality benefit from image-guided, driving-pressure, or transpulmonary pressure-based personalization [61].

### 6.2. ArtificiaI Intelligence and Closed-Loop Ventilators

Artificial intelligence (AI) and closed-loop ventilator modes are emerging as promising tools for preventing and managing VALI. Both approaches aim to deliver individualized, adaptive, and lung-protective ventilation strategies that extend beyond the limitations of conventional methods. AI has the potential to optimize and personalize mechanical ventilation by continuously integrating patient-specific physiological data to guide clinical decisions. Current research has largely focused on applications such as predicting clinical outcomes, assessing readiness for weaning, and improving extubation success [62].

Closed-loop ventilation operates through feedback mechanisms that automatically adjust ventilator outputs to match the needs of each patient. In this model, clinicians set target parameters, while the ventilator fine-tunes its settings in real time. One widely studied system, IntelliVent-ASV^®^**,** adjusts minute ventilation based on end-tidal carbon dioxide levels and modifies oxygen delivery according to peripheral oxygen saturation. In a randomized controlled trial, IntelliVent-ASV^®^ was shown to provide safe and effective ventilation [63,64].

Another promising approach is time-controlled adaptive ventilation (TCAV)**,** a closed-loop application of airway pressure release ventilation. TCAV individualizes expiratory duration based on lung compliance, aiming to stabilize alveoli and gradually reopen collapsed regions. By adapting to the dynamic pathophysiology of ARDS, TCAV may reduce VILI and improve patient outcomes [65].

### 6.3. Pharmacological Interventions Targeting Inflammation

Targeting inflammatory pathways with pharmacological therapies offers a promising approach to reducing VALI and improving patient outcomes. Damage-associated molecular patterns (DAMPs) are released during tissue injury and may contribute to VALI. DAMPs activate pattern recognition receptors that drive nuclear factor kappa-light-chain-enhancer of activated B cells (NF-κB)-mediated inflammation. Though low concentrations of DAMPs promote tissue repair, their increased release promotes lung damage, which indicates the bidirectional function of DAMPs in inflammation. Experimental data reveal that DAMP signaling is a target for anti-inflammatory inhibition and improved results [24]. The endothelial Yes-associated protein (YAP) can also be targeted pharmacologically. Through its stabilization of vascular endothelial cadherin (VE-cadherin), endothelial integrity can be sustained, which then can reduce vascular leakage and neutrophil invasion, showing its potential as a therapeutic target [66].

Multiple pharmacological agents have been shown to reduce inflammatory markers, potentially reducing the risk for VALI. Commonly used anesthetics such as isoflurane, sevoflurane, and propofol have anti-inflammatory effects via their reduction of cytokine release [27]. Similarly, remimazolam, a relatively new sedative, shows promise in targeting inflammation by limiting macrophage pyroptosis, reducing cytokine release, and preserving alveolar structure through translocator protein (TSPO) activation in preclinical studies utilizing mouse models of VILI [67]. In a meta-analysis of 20 randomized controlled trials, dexmedetomidine demonstrated anti-inflammatory effects by lowering inflammatory cytokines and improving oxygen levels in patients undergoing one-lung ventilation. The findings indicate that dexmedetomidine could be a promising anti-inflammatory option, though larger trials are needed [68]. Overall, these strategies demonstrate the potential of pharmacological treatments to reduce inflammation in VALI. However, most are still experimental and need more validation in clinical practice.

### 6.4. Stem Cell Therapies and Tissue Repair Mechanisms

Stem cell therapies, particularly mesenchymal stem cells (MSCs), are promising agents in VALI prevention and therapy by promoting regeneration, modulating inflammation, and repairing lung tissue, as evidenced by various preclinical and clinical studies. Preclinical models have demonstrated that intravenous MSC therapy can improve oxygenation and compliance, reduce edema and inflammation, and restore lung architecture in VILI rat models [69], suggesting potential therapeutic benefits for VALI. Early clinical trials have shown potential benefits of MSC therapy, such as reduced lung lesion volume in severe COVID-19 cases. In a phase 2 randomized, double-blind trial in severe COVID-19, human umbilical cord MSCs significantly reduced solid lung lesion volume compared with placebo. However, the effects on mortality and long-term function remain unclear [70]. Similarly, a pilot trial of adipose-derived MSCs in ARDS found the treatment to be safe, but it showed no significant differences in ventilator-free days, or ICU stay compared to placebo [71]. Larger trials are needed to confirm benefits, refine dosing, and determine optimal cell sources and delivery strategies [72].

## 7. Controversies and Challenges

### 7.1. Optimal PEEP Levels and Individualized Settings

The optimal level of PEEP in VALI continues to be debated. Although higher PEEP may enhance alveolar recruitment and reduce atelectrauma, it can risk alveolar overdistension, contributing to lung injury. Thus, there is growing consensus that PEEP has to be adjusted according to each patient’s lung physiology. One approach to personalization was tested in a retrospective cohort study that utilized esophageal manometry to guide PEEP settings based on transpulmonary pressure in obese, mechanically ventilated patients. This method allowed clinicians to tailor ventilation to patient-specific needs. Results of this study included lower driving pressures, and improved oxygenation, with no significant adverse events reported [73].

Electrical impedance tomography (EIT) represents another promising method for optimizing PEEP at bedside. This non-invasive, radiation-free technique provides real-time monitoring of regional ventilation [74]. A recent meta-analysis demonstrated that ARDS patients managed with EIT-guided PEEP had lower mortality compared to conventional ventilation, although no improvements in other clinical outcomes were noted [75].

### 7.2. Use of High-Frequency Oscillatory Ventilation

High-frequency oscillatory ventilation (HFOV) continues to be a topic of discussion in preventing VALI. Unlike conventional mechanical ventilation, HFOV delivers extremely small tidal volumes at very rapid respiratory rates using an oscillating piston pump. HFOV’s approach helps maintain oxygenation by sustaining higher mean airway pressures while limiting large changes in alveolar pressure, with the goal of keeping the lungs open and reducing injury [76]. HFOV was initially developed to avoid the potentially damaging volume and pressure fluctuations seen with standard ventilation. Theoretically, HFOV is particularly well-suited for surfactant-deficient infants, as it may promote more effective alveolar recruitment [22]. Clinical studies in premature infants support the benefits of HFOV since early application of HFOV has been linked to reduced cytokine-driven inflammation [77]. These findings suggest that HFOV may have meaningful benefits in carefully selected neonatal populations.

Evidence in adults, however, has been far less convincing. HFOV is primarily used as a rescue strategy in cases of severe, refractory hypoxemia, although its role in the routine management of ARDS remains uncertain [35]. Due to the limited number and relative homogeneity of adult studies, its impact on VALI and mortality in ARDS remains unproven [9]. A large multicenter trial that randomized adults with ARDS to receive either HFOV or conventional ventilation found no difference in 30-day mortality between groups, indicating that HFOV offers no clear survival advantage over standard care [78].

### 7.3. Debates Around Permissive Hypercapnia

Permissive hypercapnia has been utilized as a potential strategy in managing VALI. Low-tidal-volume ventilation, one of the cornerstones of lung-protective ventilatory strategies, often results in hypercapnia and can contribute to the development of hypercapnic acidosis, rendering permissive hypercapnia a valid option to consider [9]. Preclinical trials using rat models have shown that, together, permissive hypercapnia and lung-protective ventilation could provide protection against VALI [79]. In a single center, a cohort of 38 immunocompromised children with severe ARDS underwent a ventilatory strategy allowing high arterial partial pressure of carbon dioxide (up to 140 mmHg) to facilitate low tidal volumes and low pressures. The study concluded that this ventilatory strategy was feasible and well tolerated in a majority of cases [80]. The secondary analysis of three large prospective ARDS cohort studies also found that severe hypercapnia (arterial partial pressure of carbon dioxide ≥ 50 mmHg) was independently associated with increased ICU mortality in moderate and severe ARDS patients, proposing the potential risks of permissive hypercapnia for certain patient populations [81]. These study results suggest that, while permissive hypercapnia has protective benefits in some groups, it would potentially be harmful in others, thereby necessitating the need for prudent patient selection.

### 7.4. Ventilation in COVID-19 and Lessons Learned

At the start of the COVID-19 pandemic, some clinicians turned to non-invasive ventilation due to a concern that endotracheal intubation may worsen outcomes. Evidence was conflicting: one study reported earlier intubation had better survival [82], whereas another found higher 60-day mortality [83]. Non-invasive ventilatory strategies allowed patients with moderate respiratory insufficiency to avoid intubation and potentially reduce VALI risk, at least for a certain amount of time. This was especially important as barotrauma occurred more often in COVID-19 ARDS than in other forms of ARDS, with an incidence of about one in every six ventilated patients [84,85]. If endotracheal intubation was unavoidable, compliance with lung-protective strategies, low tidal volumes, plateau pressures below 30 cm H_2_O, and gentle titration of PEEP were the safest and most effective approaches [86].

## 8. Conclusions

Developments in critical care medicine in the last two decades have elucidated the major mechanisms of VALI, and clinical evidence has established that prevention via lung-protective ventilation strategies improves outcomes in patients with and without ARDS. Nonetheless, VALI remains a major contributor to morbidity and mortality in critically ill patients necessitating mechanical ventilation. Thus, further research is needed to shed light on optimal preventative measures, systemic effects and patient-specific susceptibility. Future directions include precision medicine approaches, integration of genomics and biomarkers, and translational research advances. Given the heterogeneity in lung mechanics, alveoli recruitability and inflammatory responses among patients with VALI, advances in precision medicine are moving away from a general to an individualized approach to medical therapeutics. Esophageal manometry, individualized dynamic PEEP titration, and EIT represent current emerging tools to tailor ventilation strategies to each patient’s physiology and pathophysiology, allowing for a potential reduction in VALI.

Advances in genomics and biomarkers of endothelial injury and inflammatory mediators may confer opportunities to identify patients at increased risk of VALI during the early injury phase, thus potentially reducing the development of VALI and improving morbidity and mortality in critically ill patients. Real-time biomarker-guided ventilation strategies may then be coupled with precision medicine to implement dynamic ventilatory measures tailored to the patient’s specific physiology. Furthermore, developments in translational research are needed to apply promising preclinical data to improved clinical outcomes. Novel technologies such as smart ventilators, AI and machine-learning-based predictive models have the potential to minimize iatrogenic lung injury while conferring ventilatory support as a life-saving measure.

This review aims to provide an integrative exploration of emerging, personalized and mechanical strategies for mitigating VALI that moves beyond conventional lung-protective ventilation measures. It promotes a paradigm shift toward precision and adaptive mechanical ventilation, incorporating patient-specific physiology, imaging and AI to dynamically tailor ventilatory strategies. Novel pharmacologic and stem cell therapies coupled with AI-guided closed-loop ventilator systems and machine-learning-based predictive models provide a framework for the prevention of VALI that incorporates molecular, inflammatory and regenerative mechanisms that move beyond purely mechanical considerations. Until these advancements are translated into sound clinical outcomes, the current strategy to prevent VALI focuses on lung-protective ventilation strategies, careful fluid balance, and adjunctive therapies such as prone positioning, use of NMBAs, and sedation minimization. Further understanding and prevention of VALI will contribute to improved outcomes in ARDS and safer mechanical ventilation measures for critically ill patients.

## Figures and Tables

**Figure 1 ijms-26-10448-f001:**
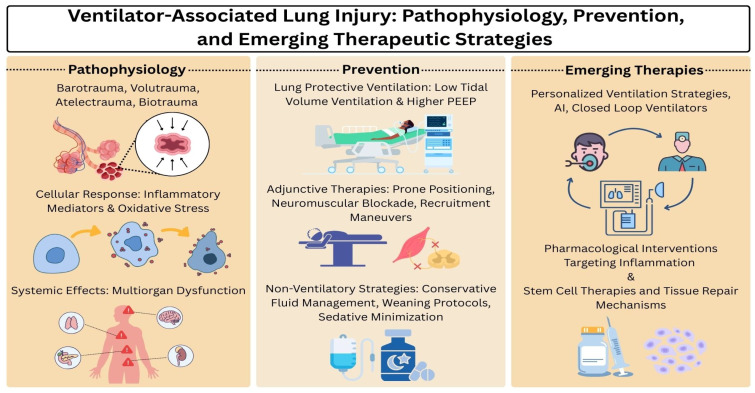
@ [Sery Pak] via Canva.com.

**Figure 2 ijms-26-10448-f002:**
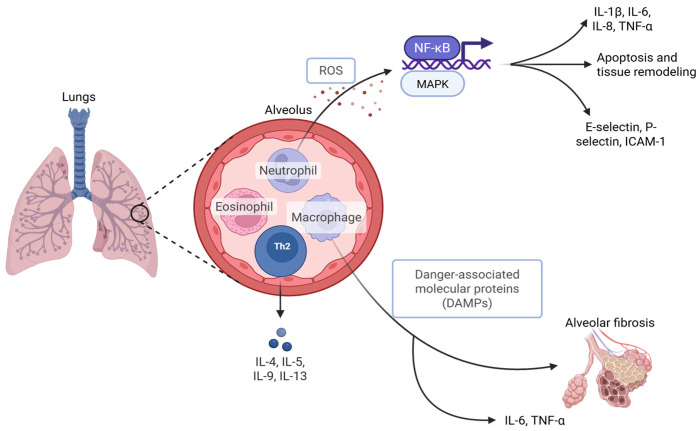
Role of immune cells in the pathophysiology of VALI. Created in BioRender. Khanyan, B. (2026) https://biorender.com/fg4zyu8 (accessed on 15 September 2025).

**Table 1 ijms-26-10448-t001:** Summary of key interventions and their demonstrated effects in the management of ARDS and prevention of VALI.

Intervention	Effect	Evidence/Source
Low tidal volume ventilation (6 mL/kg PBW, plateau ≤ 30 cm H_2_O)	Reduced mortality by ~9%, shorter duration of mechanical ventilation, lower inflammatory cytokine levels	ARDSNet (2000) [7]
Individualized PEEP titration	Improved oxygenation and reduced ventilator dependency; no consistent mortality benefit	Meade et al. [43], Mercat et al. [44]
Prone positioning (early, prolonged sessions in severe ARDS)	Improved oxygenation and significantly reduced 28- and 90-day mortality without major increase in complications	PROSEVA trial [46]
Neuromuscular blocking agents (48-h cisatracurium infusion)	Reduced 28- and 90-day mortality, fewer days on ventilation, lower incidence of barotrauma and pneumothorax	ACURASYS trial [47]
Conservative fluid management	Reduced pulmonary edema, increased ventilator-free days, improved outcomes	FACTT trial [50]
Early mobilization and sedation minimization	Improved functional recovery, increased ventilator-free days, enhanced physical and neurocognitive outcomes	Schweickert et al. [52]

## Data Availability

No new data were created or analyzed in this study. Data sharing is not applicable to this article.

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
