# Peer review of "Ventilator-Associated Lung Injury: Pathophysiology, Prevention, and Emerging Therapeutic Strategies"

_ijms, 2025, doi:10.3390/ijms262110448_

Round 1
Reviewer 1 Report
Comments and Suggestions for Authors
I read with great interest the review by Costa et al about VALI. A lot amount of work has been done, but i still have a few recommendations for the authors:
- The concept of stress and strain should be more extensively discussed.
- Mechanical power is mentioned only briefly. More details should be given and more studies shoud be discussessd.
- More references and studies should be included about driving pressure.
Author Response
Comments and Suggestions for Authors
I read with great interest the review by Costa et al about VALI. A lot amount of work has been done, but i still have a few recommendations for the authors:
- The concept of stress and strain should be more extensively discussed.
Response: We appreciate the input from the reviewer. The concepts of stress and strain have been further elucidated in the manuscript with appropriate references (Page 3 Lines 86-94).
- Mechanical power is mentioned only briefly. More details should be given and more studies should be discussed.
Response: We adopted the reviewer’s suggestion and elaborated further on the concept of mechanical power and its effects on lung injury (Page 11 Lines 427-444).
- More references and studies should be included about driving pressure.
Response: We adopted this suggestion and included more studies and their respective references on driving pressure and its effects on ventilator-induced lung injury (Page 3 Lines 89-103).
Reviewer 2 Report
Comments and Suggestions for Authors
The manuscript is overall well written. Additional focus into the molecular mechanisms involved in the "Prevention and Protective Strategies" as well as to the emerging therapies section could further align with the scope of the journal and highlight the novelty of the specific review compared to the several others that were published recently on the field.
Ventilator-Associated Lung Injury: Pathophysiology, Prevention, and Emerging Therapeutic Strategies addresses the hot topic of VILI or alternatively named VALI . While authors make an effort to combine pathophysiology, clinical data and molecular mechanisms in the current format it is overall repetitive (to existing literature) and does not add anything substantial . It has a potential to become intreresting as long as there is a detailed analysis of molecular mechanisms and more clear logical connection to clinical outcomes otherwise there are several other recent reviews covering sufficently the specific topics and with this angle of view the decision should be rejection.
Author Response
Comments and Suggestions for Authors
The manuscript is overall well written. Additional focus into the molecular mechanisms involved in the "Prevention and Protective Strategies" as well as to the emerging therapies section could further align with the scope of the journal and highlight the novelty of the specific review compared to the several others that were published recently on the field.
Response: This is an excellent suggestion by the reviewer. We have adopted this suggestion and revised our manuscript to include the novelty of our review (Page 14 Lines 605-612).
Reviewer 3 Report
Comments and Suggestions for Authors
The review article by Costa et al. deals with ventilator-associated lung injury, its pathophysiology, prevention and therapeutic strategies.
The text is logically divided into individual chapters and subchapters. The article is written at a very good professional and linguistic level. The text is written clearly, comprehensibly, the reader smoothly moves through the individual chapters.
In spite of wide use of lung-protective strategies which have significantly improved the outcome in ARDS, mortality still remains high. This article proposes the possibilities which may enhance the survival and minimize the adverse effects of ventilator treatment. Discussion on the use of new approaches is critical, considering both pros and cons.
The topic is original and relevant to the field. The article addresses a gap in the field as it both provides a review of the current therapy and reviews the perspective approaches.
This article brings a review of potentially beneficial approaches: personalised ventilation strategies, use of AI and close-loop ventilators, novel pharmacological interventions and stem cell therapies which have not been corroborated in a such big extent in other articles up to now.
I have no specific objection regarding methodology. The authors have processed the available literature in a clear and easy-to-understand manner.
Conclusions of the article are fully consistent with the evidence and arguments presented in the text and they address the posed main question: to discuss the pathophysiology of ventilator-associated lung injury, risk factors, clinical presentation and diagnosis, protective strategies, and emerging therapies.
The used references are appropriate.
In section 5, I suggest adding a review table or diagram of recommended approaches according to current guidelines. I have no additional comments on the tables and figures.
However, I have several minor comments to the language:
In the Introduction, the first paragraph: it is necessary to unify "lifesaving" and "life-saving".
There are several spelling errors in the text due to inattention, e.g. a dot instead of a comma in line 10 of the Abstract and in line 66 of the Introduction.
Author Response
Comments and Suggestions for Authors
The review article by Costa et al. deals with ventilator-associated lung injury, its pathophysiology, prevention and therapeutic strategies.
The text is logically divided into individual chapters and subchapters. The article is written at a very good professional and linguistic level. The text is written clearly, comprehensibly, the reader smoothly moves through the individual chapters.
In spite of wide use of lung-protective strategies which have significantly improved the outcome in ARDS, mortality still remains high. This article proposes the possibilities which may enhance the survival and minimize the adverse effects of ventilator treatment. Discussion on the use of new approaches is critical, considering both pros and cons.
The topic is original and relevant to the field. The article addresses a gap in the field as it both provides a review of the current therapy and reviews the perspective approaches.
This article brings a review of potentially beneficial approaches: personalised ventilation strategies, use of AI and close-loop ventilators, novel pharmacological interventions and stem cell therapies which have not been corroborated in a such big extent in other articles up to now.
I have no specific objection regarding methodology. The authors have processed the available literature in a clear and easy-to-understand manner.
Conclusions of the article are fully consistent with the evidence and arguments presented in the text and they address the posed main question: to discuss the pathophysiology of ventilator-associated lung injury, risk factors, clinical presentation and diagnosis, protective strategies, and emerging therapies.
The used references are appropriate.
In section 5, I suggest adding a review table or diagram of recommended approaches according to current guidelines. I have no additional comments on the tables and figures.
Response: We thank the reviewer for the encouraging comments and suggestions. We have incorporated a table summarizing the key interventions and their demonstrated effects in section 5 (Table 1).
However, I have several minor comments to the language:
In the Introduction, the first paragraph: it is necessary to unify "lifesaving" and "life-saving".
There are several spelling errors in the text due to inattention, e.g. a dot instead of a comma in line 10 of the Abstract and in line 66 of the Introduction.
Response: We have corrected typos as described by the reviewer and have corrected any other typos throughout the manuscript.
Round 2
Reviewer 1 Report
Comments and Suggestions for Authors
The manuscript has been substantially improved. I have no further comments.
Reviewer 2 Report
Comments and Suggestions for Authors
Authors refined certain parts of the manuscript enhancing presentation and clarifying the exact aim of the manuscript